# Associations between Fundamental Movement Skills, Physical Fitness, Motor Competency, Physical Activity, and Executive Functions in Pre-School Age Children: A Systematic Review

**DOI:** 10.3390/children9071059

**Published:** 2022-07-15

**Authors:** Chipo Malambo, Aneta Nová, Cain Clark, Martin Musálek

**Affiliations:** 1Faculty of Physical Education and Sport, Charles University, 162 52 Prague, Czech Republic; cmalambo@yahoo.com (C.M.); anetnova1@seznam.cz (A.N.); 2Centre for Intelligent Healthcare, Coventry University, Coventry CV1 5FB, UK; ad0183@coventry.ac.uk

**Keywords:** fundamental movement skill, motor competence, physical activity, physical fitness, preschool, association

## Abstract

Previous empirical research and reviews have suggested that the level of fundamental movement skills (FMS), motor competence (MC), physical activity (PA), or physical fitness seem to directly influence the executive functions (EFs) in school aged children. However, there is no available comprehensive review of whether the exact links between motor constructs and EFs also exist in the preschool period, even though preschool age is the critical period for developing EFs. Therefore, this study aimed to systematically review the evidence on the association between FMS, MC, PA, PF, and EFs. To conduct the systematic review, we utilized searches using Web of Science, PubMed, and EBSCO (including SPORTDiscus and Academic Search Premier). We included studies that examined associations between one or all of the four motor constructs with EFs among typically developing children aged 3–6 years, published between January 2010 and October 2021. A total of 15 studies met the inclusion criteria, of which four were randomized controlled trials, three were longitudinal studies, four were cohort studies, and four were cross-sectional studies. We found weak correlations or insufficient evidence for associations between FMS, PA, PF, and EFs. However, there was strong evidence for a moderately strong association between MC and working memory, a moderately weak association between MC and inhibition, and inadequate evidence for a weak to moderate association between MC and shifting. In addition, only half of the included studies were methodologically high-quality studies. Specifically, a questionable design selection of research samples might bias the strength of evaluated associations. We also found significant diversity in the diagnostic tools used for assessing and measuring motor and EFs domains. Our findings support the assumption that motor competencies level, which contains physical capacity and cognitive components, could be significantly linked to EF development from a preschool age. Therefore, we suggest that future studies focus more on clinical trial design, combining movement interventions with different levels of cognitive components, for the purposive development of EFs in preschool-aged children.

## 1. Introduction

Studies have examined the association between different motor abilities and executive functions [1,2]. Motor ability, specifically, has been defined by some authors as an inherited, relatively enduring trait of an individual that supports different kinds of cognitive activities or skills [3]. It covers task requirements, general body coordination, and well-timed movements in response to interactions with practice conditions [4,5,6]. Previous research that verified the structure of motor ability using factor analysis led to the identification of two basic motor-related factors. The first factor contained performance abilities that contribute to motor performance (dynamic strength, static strength, dynamic flexibility), and the second factor contained psychomotor abilities (response orientation, control precision, multi-limb coordination) [4,7]. These two motor abilities factors shared only a tiny portion of the variance; therefore, it was posited that motor ability manifestation is significantly determined by task specificity [8]. The measurement of these tasks among the general population or preschoolers can be noticed in different assessments that measure presumed motor abilities. Motor abilities can be observed through physical activity (PA), physical fitness level (PF), fundamental movement skills level (FMS), and motor competencies level (MC). For instance, PA, mostly measured through actigraphy, is defined as any bodily movement produced by skeletal muscles that result in energy expenditure measured in the level of intensity from sedentary to vigorous PA [9]. Physical fitness can be health- or performance-oriented and is described as participation in daily activities (or sports) without undue fatigue [10]. Some of the test batteries used to measure PF among preschoolers include the PREFIT and the EUROFIT. Fundamental movement skills, measured using the Test of Gross Motor Development (TGMD-2) battery [11], include movement such as running, leaping, catching, and kicking, which represent the essential foundations for future movement and PA, while MC means having proficiency (or competency) in FMS and can be assessed using the Motor Competence Assessment (MCA) battery [12,13]. All mentioned motor abilities are important health parameters and play a key role in human development, including the development of executive functions (EFs) [14,15,16,17]. 

Executive functions are understood as important skills for learning development; they are considered top-down control processes of human behavior, the primary role of which is to enable a person to participate in independent, goal-directed, self-serving behavior [18,19]. Diamond et al. [20] describe three basic executive functions, namely, working memory (the ability to retain and use separate pieces of information over short periods), cognitive flexibility (the ability to maintain or change attention in response to a variety of demands or to apply different rules in a variety of situations), and inhibition (the ability to prioritize tasks and avoid impulsive actions or reactions). These interact to form what are termed higher executive functions, namely, planning, problem-solving, and logical and abstract reasoning [19,21]. Psychologists propose that in humans, including children, there is a strong positive link that exists between different motor abilities and psychological benefits [22]. 

In expanding on the cognitive development of children, Piaget [23] explained that the rich interplay between biological processes of maturation and neural development of the central nervous system leads to cognitive development. This process, combined with a child’s physical maturation, and sensory-motor development, helps the child actively experience and discover the physical world [24]. Motor development in early infancy is typically seen through rudimentary movement sequences, increases in PA, and the ability to fix the stimulus of interest, all of which positively influence attention [25]. In addition, improving locomotion during infancy, mainly by bipedal walking, has been shown to be related to cognitive flexibility [26]. Therefore, early motor exploration, known as “learn to learn”, seems to influence complex EF development [27]. Furthermore, at the central nervous system (CNS) level, gross motor skills are controlled from the same regions, e.g., frontal and parietal, as EFs [28,29]. Therefore, at baseline, children with better motor skills, which may be comprised of PA, PF, FMS (the fundamental building blocks of movement), and MC (the competency with which one performs FMS), are predicted to have better attentional and preparatory processes during working memory tasks. In addition, PA enhances brain functioning in the premotor and motor cortex and the frontoparietal network, which results in better working memory [2,30]. 

Previous studies have shown that physical exercise, described as voluntary, intentional activity, also measured through PA, PF, or FMS level, causes specific biochemical and hormonal changes, such as increasing oxygen saturation and glucose delivery, improving cerebral blood flow, and increasing neurotransmitters levels, such as brain-derived neurotropic factor (BDNF) or neurotrophic growth factor (NGF) [31,32]. These changes have been shown to enhance mental and academic performance, among many other positive outcomes [1,33,34]. In pre-adolescents and adolescents, the positive effects of acute and chronic aerobic PA, PF, and MC on executive functioning and academic achievement have been documented [35,36,37,38]. However, these relationships in early childhood are unclear or inconsistent [39,40], especially regarding the degree and stability of these associations. 

Studies on preschoolers have shown that those with greater PA, FMS, PF, or MC had better working memory, attention, and/or inhibitory control [41,42,43,44,45]. Völgyi et al. [46] found that PA level in preschool-age children is strongly related to power in the alpha band (8–12 Hz) in the left and right central brain region, which relates to cognitive performance. However, in empirical studies, the strength of associations between motor ability and cognitive development has been found to be equivocal. For instance, Wen et al. [40] found no significant differences between PA, working memory, and cognitive flexibility after post-intervention. In addition, Zysset et al. [47] suggested that there are several factors that can predict EFs in preschool-age, including individual variables such as sex, social and economic status, visual perception, and fine motor skills. 

Recent literature has reviewed the relationship between PA and cognitive development in early childhood [48,49,50], albeit mostly focusing on primary school-going children and older children [51,52]. 

Van der Fels et al. [52] found no correlation in the literature or insufficient evidence for or against motor skills and cognitive skills. However, a stronger relationship between underlying motor and cognitive skills categories was found in prepubertal children, as compared to pubertal children (older than 13 years). Current evidence suggests relationships between different motor constructs and executive functions in preschool-aged children; however, we are not aware of any reviews that have comprehensively examined FMS, MC, PA, PF, and executive functions in preschool children. Development in this age group is of paramount importance. Therefore, it is vital to elucidate these relationships, particularly for effective program implementation. In addition, this would result in the development of a sound motor–cognitive model that would work as a theoretical foundation for training and interventions. Subsequently, the purpose of this review was to systematically review the evidence on the association between FMS, MC, PA, PF, and executive functions. Based on the lack of reviews on this topic, this study provides a hitherto unavailable and more comprehensive outlook. We included studies from various databases, and all studies were evaluated for methodological quality. Furthermore, we analyzed the general relationship between each motor construct and other executive functions. 

## 2. Materials and Methods

The Preferred Reporting Items for the Systematic Reviews and Meta-analysis (PRISMA) statement guidelines [53] were followed in completing this review.


**Search strategy**


The databases used for the literature search included PubMed, Web of Science, EBSCOhost, and SPORTDiscus. They were searched for records that contained one of the following combinations of terms: (#1 AND #2 AND #3 AND #4 AND #5) (1) Preschool * OR Kindergarten* OR “early childhood*” OR “young child”; (2) “motor skill” OR “Motor competence” OR “movement assessment battery” OR “fundamental motor skills” OR “physical fitness” OR “motor performance” OR “movement assessment battery” OR “fine motor development” OR “gross motor development”; (3) Physical activity* OR Physical performance*; (4) Executive function* OR Cognitive function* OR cognitive abilities; (5) Relate* OR Predict* OR effect* OR correlate* or associate* OR difference*. This review collected original articles published from January 2010 to October 2021 in English. The search strategy was applied in September 2021 and updated in November 2021. Later, we conducted a manual search of reference lists of included studies to find additional articles that might have been missed by the database search strategy. 


**Eligibility criteria**



*Research type*


We included only quantitative empirical studies (cross-sectional, longitudinal, experimental). We excluded systematic reviews and meta-analyses. 


*Area of interest*


For this review, we explored studies that investigated associations between FMS, MC, PA, PF, and EFs in children aged 3 to 6 y. The outcomes of the studies had to be objectively measured and report at least one executive function and at least one motor domain. The studies had to report a prediction, an association, correlation, and/or regression analysis between FMS and cognitive function. 


*Population*


Studies were deemed eligible if they focused on neurotypical preschool children. Studies that included neurodivergent children and/or with other health conditions were excluded. Studies with children younger than 3 or older than 6 years were not included. 


*Language*


Only studies published in English were considered for this review. 


*Selection of studies and data extraction*


The flow of the study selection process is depicted in Figure 1 [53]. Initially, duplicate articles were removed, and then the research results were screened via title and abstract using predefined inclusion and exclusion criteria. Final studies were selected after retrieval of full texts for evaluation. One reviewer screened the titles and abstracts and conducted a full-text review. Half of the articles were reviewed by an independent reviewer. In the event of disagreement between reviewers, a decision on article inclusion was attained via discussion. This process resulted in 15 articles included for analysis. 


*Assessment of methodological quality*


We used the Quality Assessment Tool for Studies with Diverse Designs (QATSDD by Sirriyeh et al. [54] to evaluate the methodological quality of the articles. This quality assessment tool has been reported to have good reliability and validity. According to standard procedure recommended by the instrument developers [54], when disagreements between the authors arose, they were discussed until issues were resolved. 

This tool has 16 items, which address both qualitative and quantitative studies. This review, however, only focused on quantitative studies; therefore, we assessed studies based on 14 items. Evaluation of the studies was based on a scale of 0 to 3 for each item, resulting in total scores ranging from 0 to a maximum score of 42. The sum obtained determined the study quality score, which was presented as a percentage of the possible maximum score [54]. The methodological quality of each paper was rated using a quartile system: 0–24% (poor), 25–50% (below average), 51–75% (average), and 76–100% (above average). 


*Data Extraction and Synthesis*


We extracted the following information from the 15 articles: the first author, year of publication, participant details, assessments and measurements of motor domains and executive function, and the study findings. Separate evaluations were made by independent authors, and the results were then compared to remove any possible inaccuracies in the information reported.

To interpret the levels of evidence, the following guidelines were used [55,56]:Strong evidence: at least three high-quality studies (quality score of 76% and above) with consistent results, where >66% find a significant association in the same direction and no more than 25% find an opposing association.Weak evidence: at least 3 average or below average quality studies with consistently positive results for or against the relationship of motor and cognitive skills.Insufficient evidence: fewer than 3 studies of whatever quality or below average or inadequate quality studies with inconsistent results for an association between motor and cognitive skills.No evidence: when there was only one study showing an association between motor and cognitive skills.

In this review, the highest quality study was one that was rated in the fourth quartile (quality score of 76% and above).

Motor skills in this review were divided into 4 domains; these included fundamental motor skills, motor competence, PA, and physical fitness. In this review, strengths of associations were reported based on the type of data analysis used and reported in each individual study. 

## 3. Results

Figure 1 shows the PRISMA flow diagram; these included the identification of literature, screening of studies, eligibility assessment, and final included articles. Accordingly, the initial search generated 137 potential records. Forty-two full-text articles were retrieved, after removing duplicates and screening titles and abstracts. Among the 42 articles, 15 studies met the inclusion criteria (4 RCTs, 3 longitudinal, 4 cohort, and 4 cross-sectional studies; Table 1).

The studies included were published between 2014 and 2021 [19,39,40,42,43,45,47,57,58,59,60,61,62,63,64]. These studies were conducted in Italy [57,60], Estonia [63], Russia [45], Switzerland [19,42,47,61], China [40], Germany [39], Canada [59], South Africa [58], Australia [43], Mexico [62], and Belgium [64]. Moreover, the sample size of participants ranged from 54 [64] to 555 [47], and only three studies [19,63,64] were longitudinal.children-09-01059-t001_Table 1Table 1General characteristics of the studies included in the systematic review.Reference (Author, Year, Country)Study DesignSample (Size (n), Age, % Girls)Motor Assessment Executive Function Assessment Outcomes **[57], Italy**RCT110 (47 girls, 63 boys), 5.23 ± 0.45 years 64 (30 girls, 34 boys), 5.18 ± 0.45 yearsEYMSCIPDARegression: Motor task a significant predictor of behavior subscale [F(1,165) = 8.61, *p* < 0.01, η2 *p* = 0.05], motor activity subscale [F(1,172) = 8.54, *p* < 0.01, η2 *p* = 0.05], linguistic comprehension subscale, [F(1,165) = 11.13, *p* < 0.01, η2 *p* = 0.06], oral expression subscale, [F(1,165) = 7.98, *p* = 0.01, η2 *p* = 0.05], metacognition subscale, [F(1,165) = 14.37, *p* < 0.01, η2 *p* = 0.08**[63], Estonia**Longitudinal 147 children (51% boys), 6.6 and 7.6 yearsActigraphPREFITEUROFITBoehm-3Regression: LPA (*p* = 0.017), MPA (*p* = 0.006), VPA (*p* = 0.011), MVPA (*p* = 0.007), and TPA (*p* = 0.012) at 6.6 years were associated with higher conceptual skills at 7.6 years.No significant associations between PA and verbal or perceptual ability.**[45], Russia**Cohort261 (boys *n* = 130, girls *n* = 131), 5.77 ± 0.32 yearsBroad jump; sit and reach test; shuttle run 4 × 5 mDCCSRegression: Inhibitory control (*p* = 0.002) and working memory (visual *p* = 0.02 and verbal *p* = 0. 03)) positively linkedwith physical fitness. Cognitive flexibility (*p* = 0.12) was not associated with PF. **[19], Switzerland**Longitudinal 134 children (68 girls and 66 boys), 6.42 ± 0.32 yearsMABC-2KTKFlanker task;Backward color recall task;Heidelberger RechentestCorrelation: PF predicts academic achievement (AA) indirectly through EFs. Significant association between PF and EFs r = 0.71 *p* < 0.05.**[40], China**RCT57 (31 boys and 26 girls), 4.40 ± 0.29 yearsActigraphFISWMSGNGSCARegression: No significant influence on inhibitory control, working memory, and cognitive flexibility with PA *p* > 0.05.**[47], Switzerland**Cohort555 (52.8% boys), 3.9 ± 0.7 yearsActigraphZNA 3-5IDS-PNEPSYRegression: PA had no effect on EFs. Fine motor skills with EFs (β = 0.17)**[61], Switzerland**Cohort156 (51% girls), 6.5 ± 4 yearsMABC-2KKTFlanker taskBackward color recall taskCorrelation: Both gross motor skills (r = 0.75) and fine motor skills (r = 0.67) correlated significantly with executive functions.**[39], Germany**RCT 101 (IG—48, 6 ± 0.43 years, 50% boys; CG—53, 6 ± 0.6 years, 45% boys)MABC-2Simon-says task; computer-based hearts-and-flowers taskCorrelation: Effect of acute coordinative exercise is temporally limited and emerges only for motor inhibition but not for cognitive inhibition or shifting.**[59], Canada**Cross-sectional95 (69.5% boys), 4.5 ± 0.7 years ActigraphTGMD-2EYTRegression: Movement behavior compositions were significantly associated with working memory (*p* = 0.01) and vocabulary (*p* = 0.00).**[58], South Africa**Cross-sectional 129 (64 urban children (47% girls), mean age 4.4 years; 65 rural children (59% girls), mean age 4.9 yearsTGMD-2EYTRegression: Inhibition (locomotor, *p* = 0.047 and object control skills, *p* = 0.02) and working memory (*p* = 0.039), but not shifting, were associated with gross motor skills. PA was not associated with inhibition and shifting but was negatively associated with working memory.**[43], Australia**RCT *n* = 111 (64 boys; Mage = 4.94 ± 0.56 years); integrated condition (*n* = 31), nonintegrated condition (*n* = 23), gesturing condition (*n* = 31), conventional condition (*n* = 26)Actigraph Free-Recall and Cued Recall TestsRegression: Children in the integrated physical exercise condition achieved the highest learning outcomes.**[60], Italy**Cross-sectional 65 children (32 boys 4.86 ± 1.04 years, and 33 girls 5.13 ± 0.89 years) MABC-2CMP, PRT; Quaiser, the Corsi block tapping test, the digitspan testCorrelation: A high correlation between two working memory tests and both mental rotation and balance was shown.**[42], Switzerland**Cross-sectional 124 children (54% girls) aged 5 to 6 years (M = 5.9 ± 0.48)MABC-2KTKOpenSesame Flanker task DCCSCorrelation: The findings demonstrate that the challenges and demands inherent in any motor task influence the magnitude of the motor–EFs link. That is, difficult (i.e., less automated) motor tasks require EFs more substantially than easy (i.e., more automated) motor tasks.**[62], Mexico**Cohort148 (56.76% boys) age 21.5 ± 3.7Peabody Motor ScaleMcCarthy ScalesRegression: Early motor performance contributes to the establishment of cognitive abilities at 5 years.**[64], Belgium**Longitudinal 54 (age kindergarten 5.98 ± 0.26; age first grade 6.95 ± 0.26 Pedometer The Dutch version of the Automated Working Memory Assessment; Flanker Task, Developmental Neuropsychological AssessmentRegression: Performance on a measure of the visuospatial sketchpad, the central executive, and fluency was predicted by children’s amount of daily PA after controlling for initial task performance.Notes: RCT: randomized control trial; EYMSC: Early Years Movement Skills Checklist; IPDA: Questionario per l’Identificazione Precoce delle Difficoltà di Apprendimento; PREFIT: Assessing FITness in PREschoolers; EUROFIT: European Physical Fitness Test Battery; Boehm-3: Boehm Test of Basic Concepts; DCCS: The Dimensional Change Card Sort; MABC-2: The Movement Assessment Battery for Children; KTK: The Körperkoordinationstest für Kinder; FIS: Flexible item selection task; WMS: working memory span task; GGG: Animal Go/NoGo task; SCA: Spatial conflict arrow task; TGMD-2: Test of gross motor development second edition; YET: Early Years Toolbox; CMP: Colored progressive matrices test; PRT: Picture rotation test.



*Methodological Quality of Studies*


Table 2 shows that a total of 7 studies [19,42,43,57,60,62,64] had high scores, whilst 8 studies [39,40,45,47,58,59,61,63] were listed as being average. Final scores were discussed among the authors, and a consensus was reached in all cases. 

For Table 2, the score range is (0–3), where 0 = not at all; 1 = very slightly; 2 = moderately; 3 = complete; the total possible score for quantitative studies is 42 and for qualitative studies is 39 [54]. 

Item 1: Explicit theoretical framework. Item 2: Statement of aims/objectives in main report. Item 3: Clear description of research setting. Item 4: Evidence of sample size considered in terms of analysis. Item 5: Representative sample of target group of a reasonable size. Item 6: Description of procedure for data collection. Item 7: Rationale for choice of data collection tool(s). Item 8: Detailed recruitment data. Item 9: Statistical assessment of reliability and validity of measurement tool(s) (quantitative studies only). Item 10: Fit between research question and method of data collection (quantitative studies only). Item 11: Fit between research question and format and content of data collection tool, e.g., interview schedule (qualitative studies only). Item 12: Fit between research question and method of analysis (quantitative studies only). Item 13: Good justification for analytic method selected. Item 14: Assessment of reliability of analytic process (qualitative studies only). Item 15: Evidence of user involvement in design. Item 16: Strengths and limitations critically discussed.


**Assessment of FMS/MC, PA, and Executive function**



*Assessment of constructs from the motor domain*


-To assess FMS, two studies used the TGMD-2 battery [58,59]. and one study used the EYMSC battery [57].-To assess motor competence, four studies used the MABC-2 [39,42,60,61], two studies used KTK [42,61], and one study each used the Peabody motor scale [62] and ZNA-3 [47].-To assess PA, five studies [40,43,47,59,63] used Actigraphs, while one study [64] used a pedometer.-To assess physical fitness, PREFIT and EUROFIT test batteries were used in two studies [45,63] and KTK [19].


*Executive function assessment*


There was a range of assessments used to measure cognitive functions, as shown in Table 1. We found that most studies assessed working memory, inhibition, shifting, and, to a lesser degree, verbal skills [43,59,62,63] and perception [63]. The most used tests were the flanker task [19,42,61,64] and the EYT [58,59]. In addition, the majority of studies used a combination of different assessments for executive functions. 


*Relationship between FMS and executive functions*


In assessing the associations between FMS and executive functions, we included three studies. Two of the studies [58,59] were cohort studies, consisting of 95 and 129 children, respectively. A study by Alesi et al. [57], however, adopted an RCT approach, with a total sample of 174. Cook et al. [58] and Kuzik et al. [59] were deemed to be of above average methodological quality, while Alesi et al. [57] was scored as average. Although Alesi et al. [57] assessed FMS with the product-oriented, EYMSC test battery, and two other studies [58,59] used the TGMD-2, which is a process-oriented test, nevertheless, only weak evidence for a strong association between FMS and working memory was demonstrated. Furthermore, Cook et al. [58] found no evidence for a strong association between FMS and inhibition [58], shifting, and verbal skills [59]. In general, the results point to weak or null evidence regarding a strong association between FMS level and performance in assessed executive functions in preschool aged children.


*Relationship between MC and executive functions*


In the MC category, we included six studies; three of the studies [47,61,62] were cohort studies, two were cross sectional [42,60], and one was an RCT [39]. Zysset et al. [47] reported the largest sample size of 555 preschoolers, with the lowest being Lehmann et al., 2014 with 65. All six articles used product-oriented tests to measure MC, where four studies [39,42,60,61] used the MABC-2, while Osorio-Valencia et al. [62] and Zysset et al. [47] used the Peabody motor scale and ZNA-3-5, respectively. There was no uniformity in tests used to measure EFs, as all articles used different measures. Only two articles [39,47] were judged to be of above average methodological quality. Table 2 and Table 3 detail strong evidence for a moderately strong association between motor competence and working memory [42,47,60,61,62]. Moreover, there was weak evidence of a moderately weak association between MC and inhibition [39,42,47,62]. We also found that there was weak evidence for a weak to moderate association between MC and shifting [39,42,47,62]. Finally, there was insufficient evidence to confirm an association between MC and verbal skills [62]. 


*Relationship between PA and executive functions*


In the category for PA, we included six articles. Two studies were longitudinal [63,64], two were RCTs [40,43], while Kuzik et al. [59] and Zysset et al. [47] were cross-sectional and cohort studies, respectively. Zysset et al. [47] had the highest sample size of 555, while Wen et al. [40] had the lowest with 57 preschoolers. All of the noted studies used accelerometers (Actigraphs) to monitor PA, except Vandenbroucke et al. [64], who utilized a pedometer. Our results indicate that there was a variety of tests used to measure EFs. Only three articles [40,47,59] had above average methodological quality, and three articles had average methodological quality. Table 2 and Table 3 show weak evidence for a weak to moderate association between PA and working memory [40,59,64]. There was also weak evidence for an association between PA and inhibition [40,47,64] and between PA and shifting [40,47,64]. Further, our results also showed weak evidence for a weak to moderate association between PA and verbal skills [43,59,63], whilst no evidence was found for a relation between PA and perception [63].


*Relationship between PF and executive functions*


Three articles were included in the category of physical fitness, where two studies were longitudinal [19,63], and one study [45] used a cohort study design. Veraksa et al. [45] had the largest sample of size of 261 preschool children, while Oberer et al. [19] and Reisberg et al. [63] reported sample sizes of 134 and 147 preschoolers, respectively. All three studies used different measures for PF, where Oberer et al. [19] used the KTK battery, Reisberg et al. [63] used both the PREFIT and EUROFIT, and Veraksa et al. [45] reported only using components of PREFIT tests. All the studies used different tests to measure EFs, and only one article [19] had an above-average methodological quality. There was insufficient evidence found for PF and working memory [19,45], PF and inhibition [19,45], and PF and shifting [19,45]. In addition, there was no evidence for an association between PF and verbal skills [63].

## 4. Discussion

This study sought to systematically review evidence pertaining to the associations between FMS, MC, PA, and PF in typically developing preschool-aged children. To our knowledge, this study is the first review to synthesis and analyze the associations between different motor skills and specific EFs in children of this age group. Accordingly, 15 studies met the eligible criteria, including *n* = 4 RCTs, *n* = 3 longitudinal studies, *n* = 4 cohort studies, and *n* = 4 cross-sectional studies. Eight studies were evaluated to be of above methodological quality, and eight were judged to be of average quality. In summary, there was weak or null evidence regarding the presence of a strong association between FMS level and the performance in assessed executive functions in preschool-aged children. Regarding the association between MC and EFs, we found strong evidence for a moderately strong association between motor competence and working memory, a moderately weak association between MC and inhibition, and weak evidence for a weak to moderate association between MC and shifting. 

In the results pertaining to the associations between PA and EFs, we found weak evidence for a weak to moderate association between PA and working memory, and weak evidence for an association between PA and inhibition and shifting. Further, the results also showed weak evidence for a weak to moderate association between PA and verbal skills, whilst no evidence was found for a relation between PA and perception. Finally, in this review, we found that there was also insufficient evidence for PF and working memory, PF and inhibition, and PF and shifting. In addition, there was no evidence for an association between PF and verbal skills. The findings in our review are essential because it is paramount to understand the specific relationships between the various motor skills (FMS, MC, PA, and PF) and specific EFs, as they may be significant for EF development and later academic achievement. 

In FMS and EFs, the associations were uncertain due to limited evidence. Indeed, there are few data in the literature that explore these relationships to elucidate a clear picture. Our study only found three studies that examined this relationship. However, it is important to note that the studies by Cook et al. [58] and Kuzik et al. [59] did show that gross motor skills were associated with better working memory, but not inhibition and cognitive flexibility. It has been demonstrated that FMS likely plays an important role in PA [65] and that attention and working memory are the first to emerge after birth [66]. It is, therefore, possible to speculate that better FMS would lead to better, more diverse PA, that would, in turn, predict better attention or working memory. However, a definitive conclusion cannot be made due to the lack of sufficient evidence currently available. 

This review showed a strong level of evidence to support a moderately strong association between motor competence and working memory. This finding is consistent with Ludyga et al. [2] and Ludyga et al. [30], where the authors mention that children with better motor skills are predicted to have better attentional and preparatory processes during working memory tasks. In trying to understand this evidence, we looked to the Stodden model, in which Stodden et al. [13] state that PA may drive the development of MC, in addition to the fact the aforementioned model posits the potential of a bi-directional relationship between MC and PA, so a causal pathway remains difficult to ascertain. Nevertheless, children with better MC might plausibly have better working memory. However, as noted in the previous literature, PA enhances brain functioning in the premotor and motor cortex and the frontoparietal network, which results in better working memory [2]. However, our results found weak evidence for a weak to moderate association between PA and working memory. We explored the evidence for the proposed hypothesis by Stodden et al. [13], and we found that Xin et al. [67] found insufficient evidence to support this hypothesis. Therefore, other factors, mainly the inclusion of PA, could be at play, as noted later in this discussion. 

Regarding PA and EFs, we found overall weak evidence for such associations. These results are consistent with previous findings that show inconsistent results. Indeed, it has been demonstrated that PA level in preschool-aged children is strongly related to the alpha band related to cognitive performance [46], while Wen et al. [40] showed no significant associations between PA and EFs post-intervention. Moreover, the effect size of the link between PA and EFs in children seems to be more significant when complex PA interventions involving strength coordination or high demand coordination motor tasks are used [68,69]. Therefore, we suggest that the inclusion of PA can significantly influence the strength of association between PA and EF. Zach and Shalom [70] pointed out that coordination (agility, manipulation) demanded PA; for example, sport games acutely increase working memory significantly more in comparison to locomotion-orientated PA such as running or walking. The concentration of neurotransmitters and brain functional concentration was also found to be higher in the high demanded coordination motor task of moderate-intensity [71]. Therefore, only information about intensity of PA may not explain accurately the link between PA and EFs. We also sought to speculate the reasons for this weak association, and using current developmental prospects, we found that several factors can purportedly influence the relationship between PA and EFs, such as biological, personal, psychosocial, and environmental (these would include parental influence and their PA habits, physical education programs) [47,72]. The mediating role of these variables needs to be considered, as they might promote or hinder how much a child participates in PA. The studies in our review did not account for most of these critical mediating variables, which might explain the inconsistency in the results. It is essential to always factor in the roles of the mediating factors in future studies to clarify the mechanism of the relationship between PA and EFs. However, this age group’s positive but weak association indicates an emerging developmental relationship. 

The associations between PF and EFs showed that there was generally insufficient evidence for these associations. One of the reasons is that we only had three studies that explored this relationship, and only one of the studies had an above-average methodological quality. Nevertheless, Veraksa et al. [45] found that inhibition, working memory, and verbal skills were positively linked to PF, while Oberer et al. [19] found that PF predicts academic achievement indirectly through EFs. However, due to insufficient evidence, it is infeasible to say if there is a relationship between these two variables. 

In trying to understand further, we identified potential factors that might influence the level of evidence found for each category and areas that future studies should incorporate. These are, namely, types of study and methodological quality, sample size, and measurements used in assessing motor skills and EFs, in addition to considering the subtle differences between FMS (the basic building blocks of movements/activities) vs. MC (one’s proficiency in conducting FMS). Firstly, in this review, only 50% of the studies had above average methodological quality, and of the 15 studies, we only found three longitudinal studies, and only two of these had above average methodological quality. The rest of the studies were either cohort, RCTs, or cross-sectional. All types of studies are important; however, more longitudinal studies on this subject are paramount to facilitate sufficient information about developmental trajectories. Such importance is reiterated by Himes [73], who mentions that growth studies provide an opportunity to yield suitable results for developing a growth reference or standard for childhood, in this case, the developmental association between motor skills and executive function. 

Next, due to the variability in this study, we found a notable range in the sample sizes used across studies. In this review, the highest number of participants (555) was in a study by Zysset [47], and the lowest (54) was in a longitudinal survey by Vandenbroucke [64]. Considering this fact, it is important to be cautious in drawing conclusions, as the different sample sizes may give different population estimates. Some studies have pointed to the importance of larger sample sizes to obtain desired nominal power. For instance, the sample size used in Vandenbroucke’s [64] longitudinal study can be defended if it is sufficient for a growth-focused study. Nevertheless, this is an area that future research should consider when designing study methods. 

Lastly, we observed wide variety used in the assessment of executive functions. Each study used one or two tools to assess executive functions, whilst most studies did not report validations for their choice of tools to assess EFs. However, it is important to remember that most of these studies were carried out in different countries, which might prove challenging regarding access to normative tools. How the heterogeneity of tests used, and the other assessment procedures, might impact outcome measures is practically challenging to discern. Thus, the question then is, how do we develop a narrowed down version of test batteries that can be used to assess EFs concerning motor skills? Hopefully, this is a matter that future studies can address. Indeed, it is important to note that the assessment of motor skills, such as FMS, MC, PA, and PF, might be influenced by the orientation of the assessment tool. It has been documented that process-oriented assessments often result in ceiling effects and floor effects, reducing validity. In contrast, product-oriented assessments do not observe the developmental movement process related to the movement product [13]. Therefore, it is important to use a multidimensional method to understand FMS, MC, PA, and PF to deepen our understanding of the relationship between these motor skills and EFs. Working memory level seems to be in preschoolers a stable EF parameter significantly associated with MC and FMS level. Surprisingly, other EF parameters, such as inhibition, shifting, verbal skills, and perception, were not strongly related to any of the selected motor constructs. One of the reasons might be the specific role of working memory, which is a predictor for other higher-order cognitive tasks, and that in pre-school age may be also the most stable part of EFs. Practically, the findings from this review could be utilized and implemented by key stakeholders, in particular pre-school care givers and parents/guardians, by considering the range of association between EFs and holistic movement/activity, such that the encouragement and facilitation of both motor and cognitively stimulating tasks should be made concurrently. Implementing a focus on developing motor and cognitive skills independently and concomitantly will likely positively influence motor and cognitive skill trajectories through childhood; of course, this is speculative, and requires empirical clarification. 

Our review has several strengths. Firstly, we analyzed the associations between different motor skills and specific EFs in preschool-aged children in a more in-depth manner than in previous studies. Secondly, we conducted a broad scope of analysis that has not been done before and analyzed four motor skills related to three executive functions; we believe that these associations are important in coming up with program interventions. Furthermore, this review had at least half of the studies included with above-average methodological quality; thus, some strong evidence for the moderate association or weak evidence for weak associations for relationships between some motor skills and EFs were identifiable. However, there is insufficient literature or evidence for or against many associations between some motor skills and executive functions. Despite some indications of associations between certain motor skills and EFs, there were not enough studies providing evidence to this effect. Notwithstanding the aforementioned strengths, this study also has limitations. Firstly, we analyzed the various skills reported in multiple studies; therefore, we did not consider that these motor skills are not necessarily mutually exclusive. Secondly, we did not conduct a meta-analysis because there was a lack of adequate studies and heterogeneity of the studies in our review. Thirdly, we solely explored the relationship between FMS, MC, PA, PF, and EFs without factoring in mediating factors, such as demographics, personal, biological, and psychosocial variables. 

## 5. Conclusions

This study systematically reviewed the associations between FMS, MC, PA, PF, and EFs in typically developing preschool-aged children. There is either no association in the current literature or insufficient evidence for or against many associations between FMS, MC, PA, PF, and EFs. However, weak to strong evidence was found for some associations, such as MC, PA, and working memory. These results suggest a complex relationship between these motor skills and EFs, where some personal, biological, and psychosocial factors could play a mediating role. However, most studies in this review did not account for most of these factors. Further, future studies need to be longitudinal to help understand these developmental trajectories more clearly. The findings from this review must be interpreted with caution due to the different study types and methodologies and methodological quality of the studies used.

Despite the lack of overarching strong evidence, we did find that there is some evidence to show that the relationship between motor skills and executive functions occurs during child development in preschool-aged children. This is interesting, because it means we can explore further to optimize motor and executive development in children through evidence-based interventions that are developmentally appropriate. Furthermore, our results suggest that highly coordination-demanding physical activities connecting physical capacity and cognitive components should be purposively involved in pre-school daily routines. We recommend that larger sample sizes be used in the future, using validated tools for EFs, and a uniform or multidimensional method for assessing motor skills. 

## Figures and Tables

**Figure 1 children-09-01059-f001:**
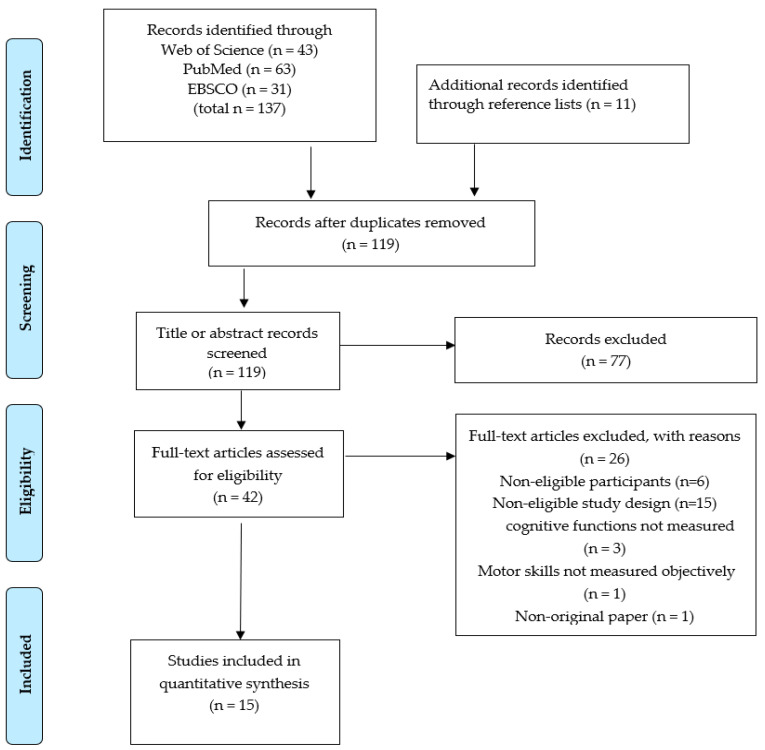
The process of article retrieval.

**Table 2 children-09-01059-t002:** Methodological quality appraisal according to the QATSDD.

Study	Criteria																	
	1	2	3	4	5	6	7	8	9	10	11	12	13	14	15	16	Score	%
**Alesi 2021**	2	3	3	1	3	3	3	1	3	2	n/a	2	2	n/a	1	1	31	74%
**Reisberg 2021**	2	1	3	2	3	3	3	2	2	2	n/a	3	3	n/a	1	2	32	76%
**Veraksa 2021**	3	3	3	2	2	2	3	2	3	2	n/a	3	3	n/a	1	0	32	76%
**Oberer 2018**	3	3	3	1	2	3	2	2	2	3	n/a	3	2	n/a	1	1	31	74%
**Wen 2018**	2	2	3	3	3	3	2	2	2	3	n/a	3	3	n/a	2	2	35	83%
**Zysset 2018**	3	3	3	1	2	3	2	2	3	3	n/a	3	3	n/a	2	1	34	81%
**Oberer 2017**	3	2	3	1	2	3	3	2	3	2	n/a	2	2	n/a	2	3	33	79%
**Stein 2017**	3	3	3	2	2	3	2	3	2	3	n/a	3	3	n/a	2	2	36	86%
**Kuzik 2020**	1	1	2	2	3	3	3	3	3	3	n/a	3	3	n/a	2	3	35	83%
**Cook 2019**	2	3	3	3	3	3	3	3	3	2	n/a	3	3	n/a	1	3	38	90%
**Mavilidi 2015**	2	3	3	2	2	3	2	3	1	1	n/a	2	1	n/a	1	2	28	67%
**Lehmann 2014**	2	3	2	1	3	3	3	2	3	2	n/a	3	2	n/a	1	0	30	71%
**Maurer 2019**	2	2	3	2	2	3	1	3	1	2	n/a	2	2	n/a	1	3	29	69%
**Osorio Valencia 2018**	2	2	3	2	2	3	1	3	1	2	n/a	2	2	n/a	2	0	27	64%
**Vandenbroucke 2016**	2	2	2	2	3	1	2	2	2	2	n/a	1	2	n/a	1	2	26	62%

Abbreviations: N/A, not applicable; QATSDD, Quality Assessment Tool for Studies with Diverse Designs.

**Table 3 children-09-01059-t003:** Summary of systematic review of the relationships between motor skills and executive functions.

Motor Skill	Cognitive Skill	No Association	Weak Association	Moderate Association	Strong Association	Evidence
Fundamental motor skills	Working-memory				[57,58,59] ^a^	Strong (strong correlation)
	Inhibition				[58]	No
	Shifting	[58]				No
	Verbal skills				[59] ^a^	No
Motor competency	Working-memory	[47] ^b^		[42]	[60,61,62]	Strong (moderately strong)
	Inhibition	[39,47] ^b^		[42]	[61]	Weak (weak moderate
	Shifting	[39,47] ^b^		[42]	[61]	Weak (weak moderate)
	Verbal skills				[62]	No
PA	Working-memory	[40,47]	[59] ^a^		[64]	Weak (weak)
	Inhibition	[40,47,64] ^b^				Weak (weak)
	Shifting	[40]			[64]	Insufficient
	Verbal skills	[63] ^c^	[59] ^b^	[43]		Weak (weak)
	Perception	[63] ^c^				No
Physical fitness	Working-memory			[61]	[45]	Insufficient
	Inhibition			[19]	[45]	Insufficient
	Shifting	[45]		[19]		Insufficient
	Verbal skills	[63] ^c^				No

^a^ Studies of Kuzik et al. (2020) reporting with the same sample size. ^b^ Studies of Zysset et al. (2018) reporting with the same sample size. ^c^ Studies of Reisberg et al. (2021) reporting with the same sample size.

## Data Availability

Not applicable.

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
