# Peer review of "Associations between Fundamental Movement Skills, Physical Fitness, Motor Competency, Physical Activity, and Executive Functions in Pre-School Age Children: A Systematic Review"

_children, 2022, doi:10.3390/children9071059_

Round 1

Reviewer 1 Report

The reviewer appreciates the opportunity to review this manuscript which presents a systematic review of the associations between motor competence, physical activity, physical fitness, movement skills, and executive functions in pre-school children.

I congratulate the authors for this paper. To the reviewer’s knowledge, there are few studies that have considered cognitive domain and their influence on the variables of the model of motor development. These aspects highlight the novelty of this manuscript.

There are, however, some comments that are aimed to improve and strengthen the manuscript in its present form.

Comment 1: Lines 61-67. Consider explain the difference between “fundamental motor skills” and “motor competence” as both can be measured by using motor competence tests (e.g., TGM, MCA), and their influence on executive functions (EFs). A non-expert reader could not see the difference and relevance to include them separately.

Comment 2: Line 90. Please, specify which are “motor skills” according to the nomenclature used in the current study. Are fundamental movement skills, motor competence or both?

Comment 3: Line 92. It was previously indicated the acronym “PA” for physical activity. I suggest authors to check the use of acronyms throughout the manuscript and amend when appropriate.

Comment 4: The results section is well-written and concise. My only suggestion is to reduce the line spacing and the letter size in table 1. It could be easy to read the results if the table is shortened.

Comment 5: Please, check the double spaces and also the absence of spacing between characters/words in the discussion (lines 305, 309, 313, 321, 327, 336, 377).

Comment 6: Can you please add a rationale or an explanation about why there are weak associations between some variables while for other seems to be a strong association. I wonder if the authors can explain/speculate why strong correlation was reported for FMS-working memory, MC-working memory while PA-working memory is weak (see Table 3). Also, why seems to be more related working memory with FMS and MC than inhibition, shifting, verbal skills or perception? For me, this part is the most important and at least the last question was not completely addressed in the discussion section. 

Comment 7: In table 3, in the “physical activity” and “working memory” row, in the first column/cell (“no association”) there are two references in brackets (Wen et al., and Zysset et al.). Please check if these references should appear written or with numbers.

Comment 8: Consider the inclusion of practical applications for pre-school teachers or agents (e.g., parents) in line with topic of the special issue.

Author Response

The reviewer appreciates the opportunity to review this manuscript which presents a systematic review of the associations between motor competence, physical activity, physical fitness, movement skills, and executive functions in pre-school children.

I congratulate the authors for this paper. To the reviewer’s knowledge, there are few studies that have considered cognitive domain and their influence on the variables of the model of motor development. These aspects highlight the novelty of this manuscript.

A; thank you to the reviewer for their kind assessment of our work. We are pleased that the novelty of our work was noted.

There are, however, some comments that are aimed to improve and strengthen the manuscript in its present form.

A: we greatly appreciate your comments, and, below, we address each point

Comment 1: Lines 61-67. Consider explain the difference between “fundamental motor skills” and “motor competence” as both can be measured by using motor competence tests (e.g., TGM, MCA), and their influence on executive functions (EFs). A non-expert reader could not see the difference and relevance to include them separately.

A: thank you for noting this. We have defined FMS as activities like running, leaping, catching, and kicking are essential foundations for future movement, whilst motor competence is the proficiency or competence with which one performs such skills. Indeed, there is a long running academic debate around the most appropriate assessment type for MC/FMS, with the argument focusing on product vs process. We feel it would be extremely difficult to briefly get into this argument (given there are whole articles dedicated just to this). However, we have included more explicit definition of FMS vs MC. (lines 61-65, and 92-95).

Comment 2: Line 90. Please, specify which are “motor skills” according to the nomenclature used in the current study. Are fundamental movement skills, motor competence or both?

A: Thank you for this comment, we have now defined what we mean by motor skills, by referring to their composition, i.e., PA, PF, FMS and MC. We have endeavored to make this clear.

Comment 3: Line 92. It was previously indicated the acronym “PA” for physical activity. I suggest authors to check the use of acronyms throughout the manuscript and amend when appropriate.

 A: edited accordingly

Comment 4: The results section is well-written and concise. My only suggestion is to reduce the line spacing and the letter size in table 1. It could be easy to read the results if the table is shortened.

 A: edited accordingly

Comment 5: Please, check the double spaces and also the absence of spacing between characters/words in the discussion (lines 305, 309, 313, 321, 327, 336, 377).

 A: edited accordingly

Comment 6: Can you please add a rationale or an explanation about why there are weak associations between some variables while for other seems to be a strong association. I wonder if the authors can explain/speculate why strong correlation was reported for FMS-working memory, MC-working memory while PA-working memory is weak (see Table 3). Also, why seems to be more related working memory with FMS and MC than inhibition, shifting, verbal skills or perception? For me, this part is the most important and at least the last question was not completely addressed in the discussion section. 

A: Thank you for this comment. We are inclined to agree with the reviewer in this regard, and have sought to include some discussion and positing around “why”. Briefly, we suggest that FMS and MC assessing contain complex motor tasks combining qualitative and quantitative aspects of motor patterns while PA can represent just simple locomotion. Furthemore, previous studies pointed on more significant link between PA and EFs’ in children (preadolescence) when in intervention the highly coordination demanding and strength coordination demanding activities were used. In addition working memory is also perceived as main predictor for other higher order cognitive functions. Therefore, in pre-school where higher cognitive functions are not even clearly present or vary a lot due to ontogeny laws the working memory might have stronger role in relation motor development. Further, it may be related to the discrepancy in the tools used to assess both MC/FMS and EFs, thus allowing error and biases to more readily appear. Indeed, this is line with our overarching conclusion that further work, considering validated tools and uniform assessments, is needed.

Comment 7: In table 3, in the “physical activity” and “working memory” row, in the first column/cell (“no association”) there are two references in brackets (Wen et al., and Zysset et al.). Please check if these references should appear written or with numbers.

A: thank you for highlighting this, we apologise for the error. This has been attributed to an error in the reference software, which has been duly rectified.

Comment 8: Consider the inclusion of practical applications for pre-school teachers or agents (e.g., parents) in line with topic of the special issue.

A: Thank you for this suggestion, we have accordingly included some practical applications in line with the special issue, “Practically, the findings from this review could be utilized and implemented by key stakeholders, in particular pre-school care givers and parents/guardians, by considering the range of association between EFs and holistic movement/activity, such that the encouragement and facilitation of both motor and cognitively stimulating tasks should be made concurrently. Implementing a focus on developing motor and cognitive skills independently and concomitantly will likely positively influence motor and cognitive skill trajectories through childhood; of course, this is speculative, and requires empirical clarification.

Reviewer 2 Report

Children

Associations between Fundamental Movement Skills, Physical Fitness, Motor Competency, 2 Physical Activity and Executive Functions in Pre-school age Children: Systematic Review

Review 6/28/22

Line 41: Studies “have”

Line 63: Comma after running.

Line 74: “prioritize” tasks

Line 231: There wasn’t a single study in the United States that met the criteria? I ask this as I have read many along these lines, but there very well could have been something that excluded them.

Line 409: Reword, fact/factors

Line 412: EFs

Lines 465-466: Reword, used the word “some” multiple times.

Check spacing in lines: 58, 101, 184, 305, 321, 327, 336, 340, 341, 346, 355, 432

There needs to be more explanation in the introduction on why FMS and MC are different. This is vaguely addressed in lines 61-64, but the rationale is not defined clearly to delineate between the two. This is important as the results are different between the two and there is discussion on these differences. A piece in the discussion to further explain the difference between the two would be beneficial as well.

Line 389: You briefly discuss the Stodden model and state that PA might drive the development of MC. Does the model also state that MC can drive PA? The explanation of the Stodden model is limited and would be very relevant to this topic to explain in more detail how the flow of the model can go both ways.

Overall Thoughts

This was a well written paper that brings value to the field and drives the fact that more research needs to be done within this area. The topic is of great value to driving intentional curriculum and the importance of physical development to cognitive processes.

Author Response

Reviewer 2

A: We thank the reviewer for their time and energy in reviewing our work, we are very appreciative and have sought to address each of your comments below.

Line 41: Studies “have”

 A: edited accordingly

Line 63: Comma after running.

  A: edited accordingly

Line 74: “prioritize” tasks

  A: edited accordingly

Line 231: There wasn’t a single study in the United States that met the criteria? I ask this as I have read many along these lines, but they very well could have been something that excluded them.

 A: thank you for noting this observation; indeed, there were many that were somewhat close to the topic at hand but did not meet all of our inclusion criteria.

Line 409: Reword, fact/factors

  A: edited accordingly

Line 412: EFs

  A: edited accordingly

Lines 465-466: Reword, used the word “some” multiple times.

  A: edited accordingly

Check spacing in lines: 58, 101, 184, 305, 321, 327, 336, 340, 341, 346, 355, 432

  A: edited accordingly

There needs to be more explanation in the introduction on why FMS and MC are different. This is vaguely addressed in lines 61-64, but the rationale is not defined clearly to delineate between the two. This is important as the results are different between the two and there is a discussion on these differences. A piece in the discussion to further explain the difference between the two would be beneficial as well.

A: thank you for this comment. Accordingly, we sought to more explicitly note the difference in the introduction (lines 61-65, and 92-95), in addition to a reiteration of this in the discussion.

Line 389: You briefly discuss the Stodden model and state that PA might drive the development of MC. Does the model also state that MC can drive PA? The explanation of the Stodden model is limited and would be very relevant to this topic to explain in more detail how the flow of the model can go both ways.

 A: Thank you for this comment, we have updated this section of the discussion to note to the potential for bidirectionality, but have not delved too far into the theoretical perspectives of the model. We elected to do this so we could continue focusing on EFs, which is not inherently present in the aforementioned model

Overall Thoughts

This was a well written paper that brings value to the field and drives the fact that more research needs to be done within this area. The topic is of great value to driving intentional curriculum and the importance of physical development to cognitive processes

A: We thank you for your appraisal of our work; indeed, we feel that this work provides a novel insight and highlights what is needed.